# SAH-Drive: A Scenario-Aware Hybrid Planner for Closed-Loop Vehicle Trajectory Generation

**Yuqi Fan** [1 2]  **Zhiyong Cui** [1 2 3]  **Zhenning Li** [4]  **Yilong Ren** [1 2]  **Haiyang Yu** [1 2]

## Abstract

Reliable planning is crucial for achieving autonomous driving. Rule-based planners are efficient but lack generalization, while learning-based planners excel in generalization yet have limitations in real-time performance and interpretability. In long-tail scenarios, these challenges make planning particularly difficult. To leverage the strengths of both rule-based and learning-based planners, we proposed the **Scenario-Aware Hybrid Planner** (SAH-Drive) for closed-loop vehicle trajectory planning. Inspired by human driving behavior, SAH-Drive combines a lightweight rule-based planner and a comprehensive learning-based planner, utilizing a dual-timescale decision neuron to determine the final trajectory. To enhance the computational efficiency and robustness of the hybrid planner, we also employed a diffusion proposal number regulator and a trajectory fusion module. The experimental results show that the proposed method significantly improves the generalization capability of the planning system, achieving state-of-the-art performance in inter-Plan, while maintaining computational efficiency without incurring substantial additional runtime.

## 1. Introduction

Autonomous driving trajectory planning primarily involves two types of algorithms: learning-based algorithms (Codevilla et al., 2018; 2019; Rhinehart et al., 2019; Zeng et al., 2019; Chitta et al., 2022; Hu et al., 2023; Jiang et al., 2023; Chen et al., 2024) and rule-based algorithms (Treiber et al.,

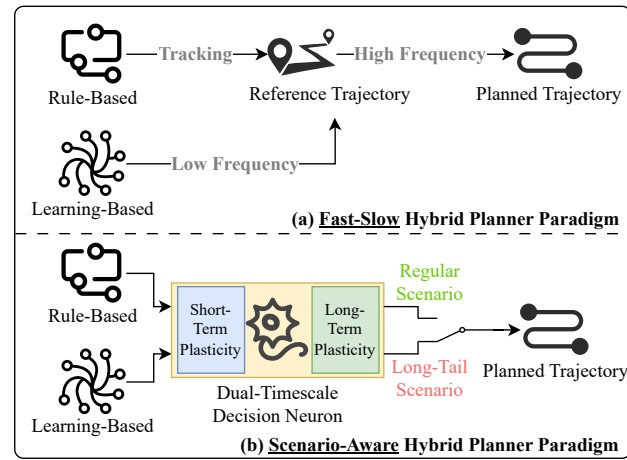

*Figure 1:* **Comparison of the fast-slow hybrid and scenario-aware hybrid planner paradigms.** (a) The learning-based planner handles low-frequency reference trajectory planning, while the rule-based planner manages high-frequency tracking. (b) The dual-timescale decision neuron enables the rule-based planner to primarily handle regular scenarios while using the learning-based planner to focus on long-tail scenarios.

2000; Fan et al., 2018; Sadat et al., 2019; Hallgarten et al., 2024). Rule-based algorithms, relying on human-crafted rules, are proficient at handling scenarios within their defined scope but are significantly limited in their ability to generalize to situations outside these predefined boundaries. Learning-based algorithms have strong generalization capabilities but typically require the construction of large models with many parameters, making real-time operation challenging and lacking interpretability. The characteristics of these two types of planners reveal that rule-based planners are well-suited for relatively simple, regular scenarios, as these scenarios constitute the majority of the driving process and can be effectively represented through predefined rules. In contrast, learning-based planners excel in handling complex long-tail scenarios (Cheng et al., 2024b), which are challenging to represent using rules and occur less frequently during driving. As such, there is a clear need for a scenario-aware hybrid planner that leverages the strengths of both

---

[1]School of Transportation Science and Engineering, Beihang University, Beijing, China [2]State Key Laboratory of Intelligent Transportation System, Beihang University, Beijing, China [3]School of Software Engineering, Beihang University, Beijing, China [4]State Key Laboratory of Internet of Things for Smart City, University of Macau, Macao, China. Correspondence to: Zhiyong Cui <zhiyongc@buaa.edu.cn>.

*Proceedings of the 42nd International Conference on Machine Learning*, Vancouver, Canada. PMLR 267, 2025. Copyright 2025 by the author(s).

rule-based and learning-based approaches, enabling the system to handle different driving situations more effectively.

**Scenario-Aware Hybrid Planner Paradigm**: Drawing from the analysis of human driving behavior and the dual-system framework (Wason & Evans, 1974; Kahneman, 2011; Leonard, 2008) underlying human cognition, it can be observed that, in regular scenarios, humans drive with little mental effort, and their behavior can be effectively captured by simple rules. In contrast, long-tail scenarios such as overtaking obstacles and passing a construction zone significantly require humans to exert more cognitive effort to identify optimal driving opportunities, requiring the sampling and evaluation of multimodal driving behaviors to make informed decisions (Mohammad et al., 2024). Thus, we first proposed the scenario-aware hybrid planner paradigm for autonomous driving trajectory planning. As shown in Figure 1, the traditional fast-slow hybrid planner paradigm (Tian et al., 2024) eliminates the distinction between scenarios to enhance the generalization ability of the planner, where the learning-based planner serves merely as guidance. The advantages of both the rule-based and learning-based planners are not effectively integrated, resulting in limited improvements in generalization. In contrast, the scenario-aware hybrid planner paradigm comprehensively combines both types of planners, enhancing generalization for long-tail scenarios while maintaining high efficiency in regular scenarios, thus offering superior performance.

In this paper, we propose the SAH-Drive based on the scenario-aware hybrid planner paradigm, with PDM-Closed (Dauner et al., 2023) as the rule-based planner and a diffusion proposal generation model as the learning-based planner. Although PDM-Closed, the State-of-the-Art (SOTA) rule-based planner, is highly efficient, it cannot perform lane changes and is not well-suited for handling long-tail scenarios (Hallgarten et al., 2024). Therefore, we train a multimodal diffusion-based proposal generator as a "second brain" to complement the rule-based planner. Additionally, we introduce a proposal number regulator that dynamically adjusts the number of generated proposals based on the highest-scoring trajectory, evaluated using predefined performance metrics (Dauner et al., 2023). This increases the diversity of proposals while reducing redundancy and accelerating the trajectory planning process.

To mimic human decision-making behavior, a dual-timescale decision neuron is designed to manage the transition between the two planners. Inspired by biological neurons, it exhibits both short-term and long-term plasticity: Short-term plasticity, which enables rapid responses to environmental changes, is governed by score-based and scenario-based rules. Long-term plasticity, which allows the system to retain and reflect on past planner performance, is achieved through a decision neuron based on Spike-Timing

Dependent Plasticity (STDP). This biologically inspired mechanism is used independently, without relying on a full spiking neural network (SNN) framework. The advantage of this approach lies in its ability to flexibly adjust decisions based on changes in planner performance by simulating the mechanisms of human neurons, enabling scene-aware planning period switching. The code for this paper is available at https://github.com/richie-live/SAH-Drive.

Our contributions are as follows:

- We proposed a **scenario-aware hybrid planner paradigm** that integrates rule-based and learning-based trajectory planning methods, and designed a dual-timescale decision neuron which is composed of the score-based switching rule, the scenario-based switching rule, and the STDP-based decision neuron.

- We integrate the SOTA rule-based planner PDM-closed and a diffusion proposal generation model into **SAH-Drive**, which significantly reduces the required training data compared to fully learning-based planners, while maintaining strong planning performance, particularly in long-tail scenarios.

- We employ a **real-time proposal number regulator** and a **trajectory fusion module** to accelerate trajectory planning while ensuring the safety of the learning-based planner. The experimental results demonstrate the efficiency and robustness of the proposed method.

## 2. Related Work

### 2.1. Rule-Based V.S. Learning-Based Trajectory Planning Methods

**Rule-based** planners use a set of predefined, hard-coded rules and heuristics to generate trajectories based on the vehicle's environment and constraints. A typical rule-based planner is the Intelligent Driver Model (IDM) (Treiber et al., 2000), which calculates the ego-vehicle's acceleration along a path based on its speed, the distance to the vehicle ahead, and predefined parameters to ensure safe and efficient driving behavior. The recently proposed PDM-Closed (Dauner et al., 2023) is an extension of IDM, which uses IDM to plan proposals, performs simulations and scoring, and ultimately selects the best trajectory. It achieved SOTA performance for rule-based planners on the nuPlan dataset.

**Learning-based** planners leverage machine learning models, typically trained on large datasets, to predict and generate optimal trajectories based on observed patterns and past experiences. End-to-end methods such as UniAD (Hu et al., 2023), VAD (Jiang et al., 2023), and VAD2 (Chen et al., 2024), as well as imitation learning-based methods like PlanTF (Cheng et al., 2024b), Pluto (Cheng et al., 2024a),

and BeTop (Liu et al., 2024), enhance the generalization of planners by incorporating neural networks into either specific modules or the entire framework. Additionally, a reinforcement learning-based method called CarPlanner (Zhang et al., 2025) has been recently proposed, which leverages temporal consistency and expert-guided rewards to generate consistent multi-modal trajectories.

Recent efforts in the autonomous driving community have also attempted to combine the rule-based planner with the learning-based planner, in order to create a **dual planner** that maintains the efficiency of rule-based planners while also achieving the generalization capabilities of learning-based planners. This concept has been applied in both DriveVLM-Dual (Tian et al., 2024) and DualAD (Wang et al., 2024), with corresponding experimental validation. The scenario-aware hybrid planner paradigm proposed in this paper is also inspired by this idea.

### 2.2. Diffusion Model for Trajectory Generation

Diffusion models can model data distributions and generate data similar to the training data, which have shown exceptional generative capabilities on various tasks, such as image generation (Song et al., 2020) and text generation (Austin et al., 2021). Furthermore, diffusion models can model the distribution of trajectories and be used for trajectory generation, which has been the focus of much research recently.

**Diffusion model in robotics**: Diffuser (Janner et al., 2022) first used a diffusion model to integrate the trajectory planning into the model sampling process by iteratively denoising trajectories, and implemented reinforcement learning counterparts to classifier-guided sampling and image in-painting. Decision Diffuser (Ajay et al., 2022) proposed modeling policies as return-conditional diffusion models within the framework of conditional generative modeling, further improving Diffuser's performance.

**Diffusion model in autonomous driving trajectory planning**: STR (Sun et al., 2023) used a diffusion-based key point decoder to model the multimodal distribution of future states resulting from interactions among multiple road users. Diffusion-ES (Yang et al., 2024) first combined the diffusion model with an evolutionary search algorithm and used it in the autonomous driving trajectory planning task. The diffusion model is used to learn the diverse trajectory distribution from the nuPlan dataset and generate similar trajectories. These generated trajectories are then used as an initial population for evolutionary search, iterating towards the optimal trajectory. This method integrates multimodal information, but due to the use of evolutionary search, it is slower in planning and not suitable for real-time operation. More recently, approaches like DiffusionDrive (Liao et al., 2024) and Diffusion Planner (Zheng et al., 2025) extend diffusion models for planning by introducing truncation around

anchors and flexible guidance mechanisms, respectively.

## 3. Method

### 3.1. Diffusion Proposal Generation Model

As demonstrated in the study of Diffusion-ES (Yang et al., 2024), reducing the amount of conditioning information broadens the distribution for trajectory sampling, thereby enhancing generalization to out-of-distribution (OOD) tasks. Hence, we adopt a non-conditional diffusion model, primarily referencing Diffusion-ES, removing evolutionary search and enhancing the trajectory encoder to design the proposal generation model. The diffusion probabilistic model for vehicle trajectory generation is detailed in Appendices A and B, which is composed of three key stages: **Feature Fusion**, **Self-Attention Fusion**, and **Decoding and Denoising**.

### 3.2. Overall Architecture of SAH-Drive

The overall architecture of SAH-Drive, as shown in Figure 2, consists of three main components: trajectory candidates generation, trajectory scoring, and dual planner switching. Initially, the lane centerline from the starting point to the endpoint is generated through Dijkstra search, followed by the PDM proposal generator and diffusion proposal generator, which produce PDM proposals and diffusion proposals, respectively. To satisfy the physical constraints of the trajectory, these proposals are evaluated through simulation using an LQR controller, producing PDM trajectories and diffusion trajectories. These trajectories are then evaluated using the widely adopted PDM score (Dauner et al., 2023), with further details provided in Appendix D. The planner switching is performed based on the dual-timescale decision neuron to select the final output trajectory. The learning-based planner operates based on a diffusion model with a lower frequency, while the rule-based planner runs based on PDM-Closed with a higher frequency. During each execution of the learning-based planner, the dual-timescale decision neuron determines the better planner.

It is worth noting that SAH-Drive uses a diffusion model as the learning-based planner to sample multimodal proposals. In fact, it can also be replaced with other existing learning-based planners such as PlanTF (Cheng et al., 2024b) and Pluto (Cheng et al., 2024a). Following the SAH paradigm, we integrated these planners with PDM-Closed as well. The corresponding experimental results are provided in Appendix F.

**Proposal Number Regulator**: To improve planning efficiency, we implemented a dynamic proposal number regulator that adaptively adjusts the number of diffusion proposals in real-time based on the highest diffusion trajectory score.

Specifically, the number of diffusion trajectories $N'$ is dy-

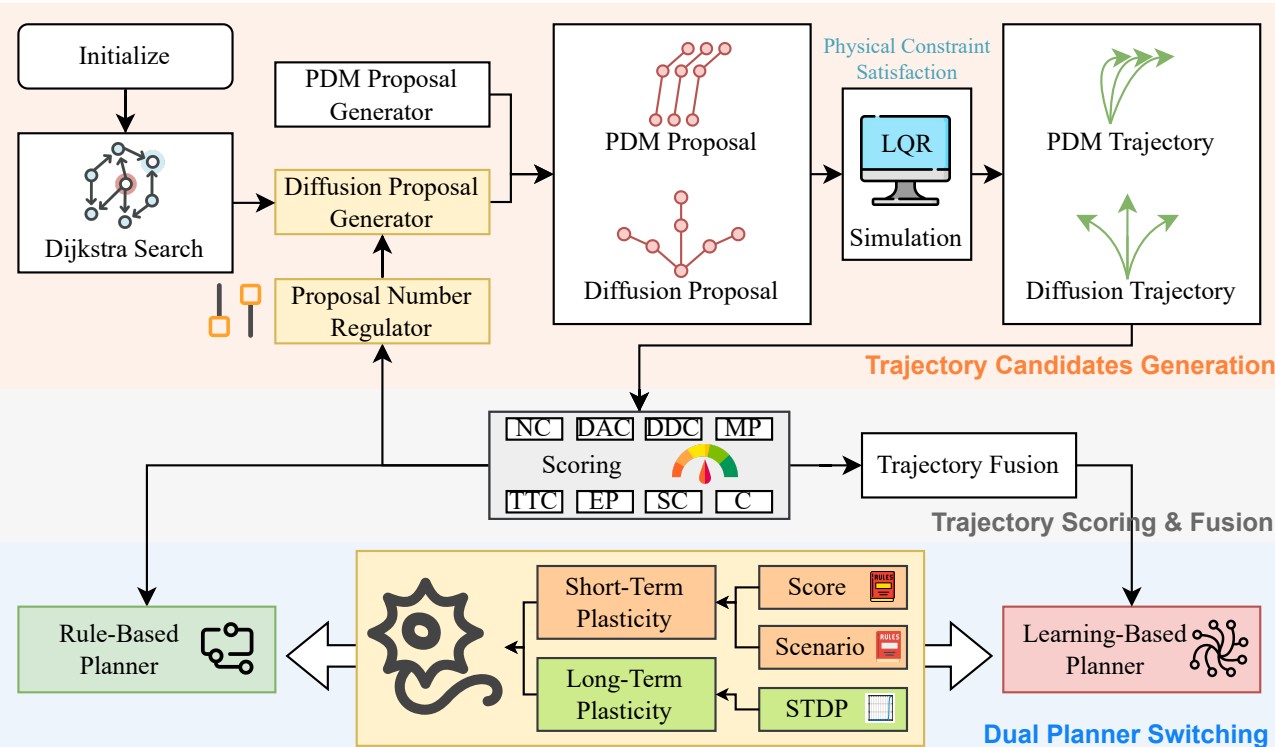

*Figure 2:* **The overall architecture of SAH-Drive**. (a) Given the starting point, endpoint, and map information, dynamically generate the PDM and diffusion proposals, then convert them into trajectories. (b) Evaluate and fuse the trajectories based on No at-fault Collisions (NC), Drivable Area Compliance (DAC), Driving Direction Compliance (DDC), Making Progress (MP), Time to Collision (TTC), Ego Progress (EP), Speed-limit Compliance (SC), and Comfort (C). (c) Switch between the rule-based planner and the learning-based planner based on the dual-timescale decision neuron.

namically adjusted based on the highest diffusion trajectory score $s_{\text{diff}}$ relative to a threshold $\tau$. The trajectory count is updated based on the highest trajectory score:

$$N' = \begin{cases} \frac{N}{2}, & s_{\text{diff}} > \tau, \\ 2N, & s_{\text{diff}} < \tau. \end{cases} \quad (1)$$

To regulate trajectory count, we apply:

$$N' = \max(N_{\min}, \min(N', N_{\max})) \quad (2)$$

Because the rule-based planner serves as a fallback, the reduction in the number of diffusion proposals has little impact on the planning performance.

**Trajectory Fusion for the Learning-Based Planner**: To mitigate the risk of excessively aggressive diffusion trajectories, which could pose high driving risks to the ego vehicle, we propose a novel fusion approach that combines the highest-scoring diffusion trajectory with the highest-scoring PDM trajectory. Given that the fused trajectory may not inherently adhere to the physical constraints of the vehicle, we first identify the proposals corresponding to the highest-scoring diffusion and PDM trajectories. The fused proposal,

$p_{\text{fused}}$, is then calculated through an exponential weighting mechanism, as shown in Equation (3):

$$p_{\text{fused}} = \frac{e^{\alpha(s_{\text{PDM}} - s_{\max})} p_{\text{PDM}} + e^{\alpha(s_{\text{diff}} - s_{\max})} p_{\text{diff}}}{e^{\alpha(s_{\text{PDM}} - s_{\max})} + e^{\alpha(s_{\text{diff}} - s_{\max})}} \quad (3)$$

where $p_{\text{PDM}}$ and $p_{\text{diff}}$ represent the PDM and diffusion trajectories, respectively, while $s_{\text{PDM}}$ and $s_{\text{diff}}$ denote their corresponding scores. The parameter $\alpha$ controls the sensitivity of the weighting to the trajectory scores. The term $s_{\max} = \max(s_{\text{PDM}}, s_{\text{diff}})$ is introduced to enhance numerical stability, preventing overflow during exponential calculations. The reason for applying exponential weighting to the trajectories is to assign significantly higher weights to the trajectory with a higher score (Li et al., 2024).

### 3.3. Dual-Timescale Decision Neuron

Neurons, as the core of human intelligence, exhibit both short-term and long-term plasticity. Short-term plasticity refers to the transient changes in synaptic strength, enabling neurons to quickly adjust and respond to immediate stimuli, thereby rapidly adapting to environmental changes. Long-term plasticity allows neurons to form lasting changes in

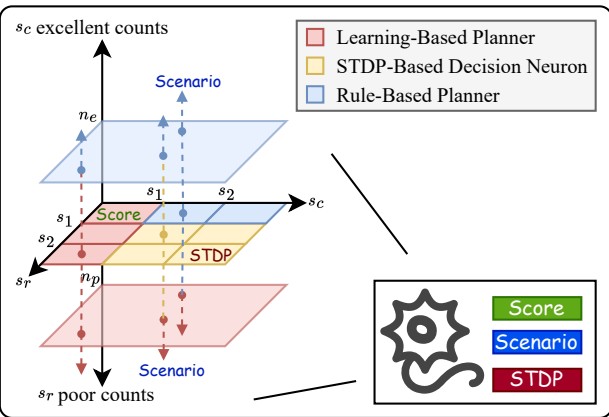

*Figure 3:* **The decision space of the dual-timescale decision neuron**. In the horizontal dimension, the red and blue areas represent the score-based switching rule, the yellow area represents the STDP-based decision neuron that selects the planner, and the vertical dimension represents the scenario-based switching rule.

their connections, which is crucial for learning and memory. Therefore, we designed the score-based switching rule and the scenario-based switching rule as the short-term plasticity of the dual-timescale decision neuron, enabling immediate transitions between planners according to their performance. Additionally, we implemented an STDP-based decision neuron as the long-term plasticity of the dual-timescale decision neuron, allowing for the memory and reflection of the planners' performance. The three-dimensional decision space for the dual-timescale decision neuron is shown in Figure 3, and its algorithmic form is presented in Algorithm 1.

**Score-Based Switching Rule**: Let the score of the rule-based planner be denoted as $s_c$ and the score of the learning-based planner as $s_r$. Based on two threshold values, $s_1$ and $s_2$, both $s_c$ and $s_r$ are classified into three levels: excellent, ordinary, and poor. When both $s_c$ and $s_r$ are poor, the learning-based planner is selected to seek opportunities. When one planner performs poorly while the other is at an ordinary or excellent level, the ordinary or excellent planner is chosen. When both $s_c$ and $s_r$ are at an ordinary or excellent level, the STDP-based decision neuron is activated to select the better planner.

**Scenario-Based Switching Rule**: The system should revert to the rule-based planner after invoking the learning-based planner, for the sake of efficient planning. If the best PDM score maintains excellent for $n_e$ consecutive runs, as shown in Figure 3, it indicates that the rule-based planner is performing well in the current scenario and should be used. During experiments, we observed that if the learning-based planner consistently produces low scores over $n_p$ iterations, it tends to remain in low-scoring states in subsequent plan-

---

**Algorithm 1** Planner Selection Using Dual-Timescale Decision Neuron

**Require:** Rule-based planner score $s_c$, Learning-based planner score $s_r$
**Ensure:** Selected planner
  1: Update weights $w_r$ and $w_c$ using Equation (5)
  2: Update consecutive counts $n_e$ and $n_p$
  3: category $\leftarrow$ decision_space($s_r, s_c, n_e, n_p$)
  4: **if** category $=$ score **then**
  5:     planner $\leftarrow$ score_rule($s_r, s_c$)
  6: **else if** category $=$ scenario **then**
  7:     planner $\leftarrow$ scenario_rule($n_e, n_p$)
  8: **else**
  9:     planner $\leftarrow$ STDP_neuron($w_r, w_c$)
 10: **end if**
 11: **return** planner

---

ning frames. This typically occurs in scenarios where the vehicle becomes stuck behind another vehicle and cannot overtake, resembling a local minimum. We refer to this state as trapped and use it as one of the entry points for the learning-based planner, where the system continuously searches for opportunities to escape the local minimum.

**STDP-Based Decision Neuron**: To facilitate memory and reflection on the performance of the dual planner during the planning process, we draw on the concept of synaptic plasticity from neuroscience. Synaptic plasticity can be simply understood as the process of adjusting the "bridges" between neurons. By modifying the width or strength of these "bridges", neural networks can autonomously regulate the speed and efficiency of information transmission, thereby improving learning and memory processes. **Spike-Timing Dependent Plasticity (STDP)** (Markram et al., 2011) is a typical synaptic plasticity mechanism where synaptic weight depends on the timing of presynaptic and postsynaptic spikes. Specifically, if the presynaptic neuron fires a spike before the postsynaptic neuron, the synaptic weight will increase; if the postsynaptic neuron fires a spike before the presynaptic neuron, the synaptic weight will decrease. This mechanism can be mathematically described by the following formula:

$$\Delta w = \begin{cases} A^+ \cdot \exp\left(\frac{-\Delta t}{\tau^+}\right) & \text{if } \Delta t > 0 \quad \text{(LTP)} \\ -A^- \cdot \exp\left(\frac{\Delta t}{\tau^-}\right) & \text{if } \Delta t < 0 \quad \text{(LTD)} \end{cases} \quad (4)$$

where $\Delta t = t_{\text{post}} - t_{\text{pre}}$ is the time difference between the firing times of the postsynaptic and presynaptic neurons, $A^+$ and $A^-$ are the gain factors for long-term potentiation (LTP) and long-term depression (LTD), respectively (usually $A^+ > A^-$), and $\tau^+$ and $\tau^-$ are the time constants associated with LTP and LTD, controlling the timescale of synaptic weight changes. Specifically, for a positive time difference ($\Delta t > 0$), when the presynaptic neuron fires be-

fore the postsynaptic neuron, LTP occurs, and the synaptic weight increases. For a negative time difference ($\Delta t < 0$), when the postsynaptic neuron fires before the presynaptic neuron, LTD occurs, and the synaptic weight decreases.

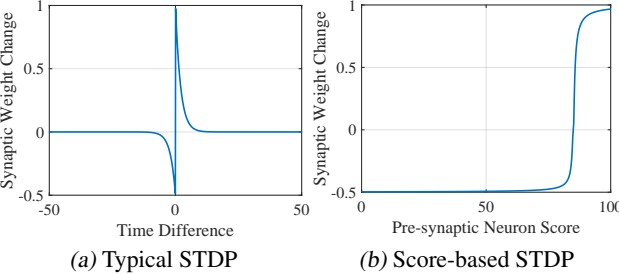

*(a)* Typical STDP        *(b)* Score-based STDP

*Figure 4:* **Illustration graph of STDP**. (a) The x-axis represents the time difference between the activation of the two neurons, while the y-axis represents the change in the connection weight between the two neurons. (b) The x-axis represents the planner's score, while the y-axis shows the change in the connection weight between the planner and the decision neuron.

Based on the STDP, we can design a decision neuron to switch between the rule-based planner and the learning-based planner. Specifically, the planners are treated as presynaptic neurons, and a decision neuron is designed as the postsynaptic neuron. The synaptic weights between the presynaptic and postsynaptic neurons are updated using the STDP mechanism. Mimicking the STDP formula, the score of the neuron (planner) is treated as the spike time. Unlike typical STDP, where the magnitude of synaptic weight change increases as the absolute time difference decreases, in score-based STDP, larger absolute score differences lead to greater synaptic changes. Thus, we use the negative reciprocal of the score difference as the time difference: $\Delta t = \frac{-1}{s_{\text{post}} - s_{\text{pre}}}$. The weight update follows the formula:

$$\Delta w = \begin{cases} A^{+} \cdot e^{-\frac{1}{(s_{\text{pre}} - s_{\text{post}})\tau^{+}}} & \text{if } s_{\text{post}} < s_{\text{pre}} \quad \text{(LTP)} \\ -A^{-} \cdot e^{\frac{1}{(s_{\text{pre}} - s_{\text{post}})\tau^{-}}} & \text{if } s_{\text{pre}} < s_{\text{post}} \quad \text{(LTD)} \end{cases}$$
(5)

where $s_{\text{pre}}$ is the score of the presynaptic neuron (the score of the planner), and $s_{\text{post}}$ is the score of the postsynaptic neuron, which is set as the threshold value for the excellent score $s_1$. The illustration graphs of the typical STDP and the proposed score-based STDP are shown in the Figure 4.

## 4. Experiment

In this section, SAH-Drive is assessed within the nuPlan benchmark (Karnchanachari et al., 2024), a well-recognized framework that incorporates estimated perception data for vehicles, pedestrians, lanes, and traffic signs. We aim to an-

swer the following research questions: 1) Does SAH-Drive exhibit scenario-aware characteristics by employing the rule-based planner in regular scenarios and the learning-based planner in long-tail scenarios? 2) Can better planning performance be achieved by combining the rule-based planner and the learning-based planner?

### 4.1. Datasets and Metrics

**Training dataset**: The nuPlan Mini dataset is a compact version of the full nuPlan dataset, designed for efficient experimentation in autonomous driving. Despite its reduced size, it retains sufficient diversity for tasks like detection, prediction, and planning, making it ideal for rapid prototyping and algorithm validation. **Validation dataset**: We use Val14 (Dauner et al., 2023) and Test14-Random (Cheng et al., 2024b) to evaluate the planner's performance in regular scenarios and interPlan (Hallgarten et al., 2024) and Test14-Hard (Cheng et al., 2024b) to assess its performance in long-tail scenarios.

**Metrics**: As the proposed planner is not an imitation-based method, open-loop evaluation metrics are not the main focus. Therefore, the evaluation is conducted using the closed-loop score reactive (CLS-R) and closed-loop score non-Reactive (CLS-NR) metrics (Karnchanachari et al., 2024). The baselines are listed in Appendix C.

### 4.2. Quantitative Results

We conducted a simulation analysis comparing the proposed method with baselines. The planner's performance on interPlan is shown in Table 1, and its performance on four nuPlan splits, including both regular and long-tail scenarios, is presented in Table 2, from which we derive the following findings:

1. PDM-Closed exhibits **conservative effects**, while the diffusion proposal generator demonstrates **radical effects**. PDM-Closed performs better in the jaywalker scenario but has lower scores in the overtaking and construction zone scenarios. On the other hand, Diffusion-ES, by continuously seeking opportunities through the diffusion model, excels at lane-changing, achieving good performance in overtaking but scoring lower in the jaywalker scenario, due to collisions with pedestrians that appear suddenly. This is consistent with human driving experience: The jaywalking scenario indeed requires conservative driving, while the overtaking scenario necessitates radical driving.

2. The proposed method achieves the highest score in inter-Plan and Test14-Hard, and delivers near SOTA performance on Val14 and Test14-Random, validating its **effectiveness in both regular and long-tail scenarios**. Since the planner in this paper is a flexible combination of a diffusion model and PDM-Closed, the comparison between SAH-

*Table 1:* **Specific comparison with SOTA planners on interPlan**. interPlan includes eight types of long-tail scenarios: construction zones, accident zones, jaywalkers, nudging, overtaking, low traffic density lane-changing, medium traffic density lane-changing, and high traffic density lane-changing. The highest score in each scenario is indicated in **bold**, while the second-highest score is underlined.

| | Planner | Type | interPlan | Constr. | Acc. | Jayw. | Nudge | Overt. | LTD | MTD | HTD |
|---|---|---|---|---|---|---|---|---|---|---|---|
| SOTA | PDM-Closed (CoRL 2023) | Rule | 42 | 18 | 0 | 48 | 74 | 9 | 62 | 62 | 62 |
| | STR2 (arxiv 2024) | Learning | 46 | / | / | / | / | / | / | / | / |
| | HybridLLMPlanner (IROS 2024) | Hybrid | 53 | 27 | 20 | 48 | **93** | 28 | **81** | 48 | **80** |
| | Diffusion-ES (CVPR 2024) | Learning | 57 | 71 | **51** | 13 | 88 | 52 | 61 | 58 | 61 |
| | PlanTF (ICRA 2024) | Learning | 33 | 9 | 0 | 33 | 49 | 9 | 50 | 40 | 73 |
| | Pluto (arxiv 2024) | Learning | 48 | 54 | 9 | 56 | 82 | 17 | 47 | 47 | 68 |
| | Diffusion Planner (ICLR 2025) | Learning | 24 | 17 | 0 | 7 | 70 | 15 | 41 | 22 | 17 |
| | SAH-Drive (Ours) | Hybrid | **64** | **72** | 44 | 47 | 80 | **78** | 64 | **63** | 63 |
| Suboptimal | Urban Driver (CoRL 2022) | Learning | 4 | 0 | 0 | 0 | 0 | 0 | 0 | 29 | 0 |
| | GameFormer (ICCV 2023) | Learning | 11 | 0 | 0 | 48 | 0 | 0 | 0 | 20 | 21 |
| | DTPP (ICRA 2024) | Learning | 25 | 18 | 18 | 44 | 10 | 0 | 40 | 36 | 34 |
| | IDM (Phys. Rev. E) | Rule | 31 | 0 | 0 | **66** | 0 | 0 | 61 | 61 | 61 |

*Table 2:* **Comparison with SOTA planners on different splits of nuPlan**. Including interPlan (long tail), Val14 (regular), Test14-Random (regular), Test14-Hard (long tail). The highest score is indicated in **bold**, while the second-highest score is underlined.

| | Planner | Type | interPlan | Val14 (R) | Val14 (NR) | Test14-Random (R) | Test14-Random (NR) | Test14-Hard (R) | Test14-Hard (NR) |
|---|---|---|---|---|---|---|---|---|---|
| SOTA | PDM-Closed | Rule | 42 | 92 | **93** | **91** | **90** | 75 | 65 |
| | STR2 | Learning | 46 | **93** | / | / | / | / | / |
| | HybridLLMPlanner | Hybrid | 53 | / | / | / | / | / | / |
| | Diffusion-ES | Learning | 57 | 92 | / | / | / | 77 | 77 |
| | PlanTF | Learning | 33 | 77 | 84 | 80 | 85 | 61 | 69 |
| | Pluto | Learning | 48 | 78 | 89 | 78 | 89 | 60 | 70 |
| | DiffusionPlanner | Learning | 24 | 83 | 90 | 83 | 89 | 69 | 75 |
| | SAH-Drive | Hybrid | **64** | 90 | 89 | 87 | 86 | **83** | **78** |
| Suboptimal | UrbanDriver | Learning | 4 | 50 | 69 | 67 | 52 | 49 | 50 |
| | GameFormer | Learning | 11 | 75 | 80 | 82 | 84 | 67 | 68 |
| | DTPP | Learning | 25 | 73 | / | / | / | / | / |
| | IDM | Rule | 31 | 77 | 75 | 74 | 70 | 62 | 56 |

*Table 3:* **Ablation experiments on the planner hierarchy and the dual-timescale decision neuron hierarchy**. The percentage following the score of each variant indicates the score change compared to the original SAH-Drive.

| Variant | interPlan | Constr. | Acc. | Jayw. | Nudge | Overt. | LTD | MTD | HTD |
|---|---|---|---|---|---|---|---|---|---|
| Original | **64** | **72** | **44** | **47** | **80** | **78** | **64** | **63** | **63** |
| Learning-Based | 62(-3.13%) | 79(9.72%) | 44(0.00%) | 44(-6.38%) | 79(-1.25%) | 74(-5.13%) | 59(-7.81%) | 63(0.00%) | 57(-9.52%) |
| Rule-Based | 40(-37.50%) | 18(-75.00%) | 0(-100.00%) | 29(-38.30%) | 75(-6.25%) | 9(-88.46%) | 62(-3.13%) | 62(-1.59%) | 63(0.00%) |
| Score Rule | 58(-9.38%) | 60(-16.67%) | 26(-40.91%) | 46(-2.13%) | 80(0.00%) | 67(-14.10%) | 62(-3.13%) | 62(-1.59%) | 62(-1.59%) |
| Scenario Rule | 43(-32.81%) | 25(-65.28%) | 5(-88.64%) | 29(-38.30%) | 75(-6.25%) | 24(-69.23%) | 63(-1.56%) | 62(-1.59%) | 63(0.00%) |
| Decision Neuron | 62(-3.13%) | 62(-13.89%) | 53(20.45%) | 47(0.00%) | 80(0.00%) | 68(-12.82%) | 63(-1.56%) | 62(-1.59%) | 63(0.00%) |

Drive, PDM-Closed, and Diffusion-ES demonstrates that SAH-Drive combines the advantages of both rule-based and learning-based planners. This is accomplished by utilizing a dual-timescale decision neuron, thereby achieving high performance in both jaywalking and overtaking scenarios.

3. The use of a scenario-aware hybrid planner paradigm significantly **reduces the data requirements** for the learning-based planner and **allows for the use of simpler structures**. The diffusion model in this study was trained only on the nuPlan mini dataset, yet it outperforms models trained on the entire nuPlan dataset, as most of the planning knowledge has already been internalized by the rule-based planner. Comparisons with the HybridLLMPlanner also suggest that training a dedicated generative model for autonomous driving may be a better choice than enhancing traditional mo-

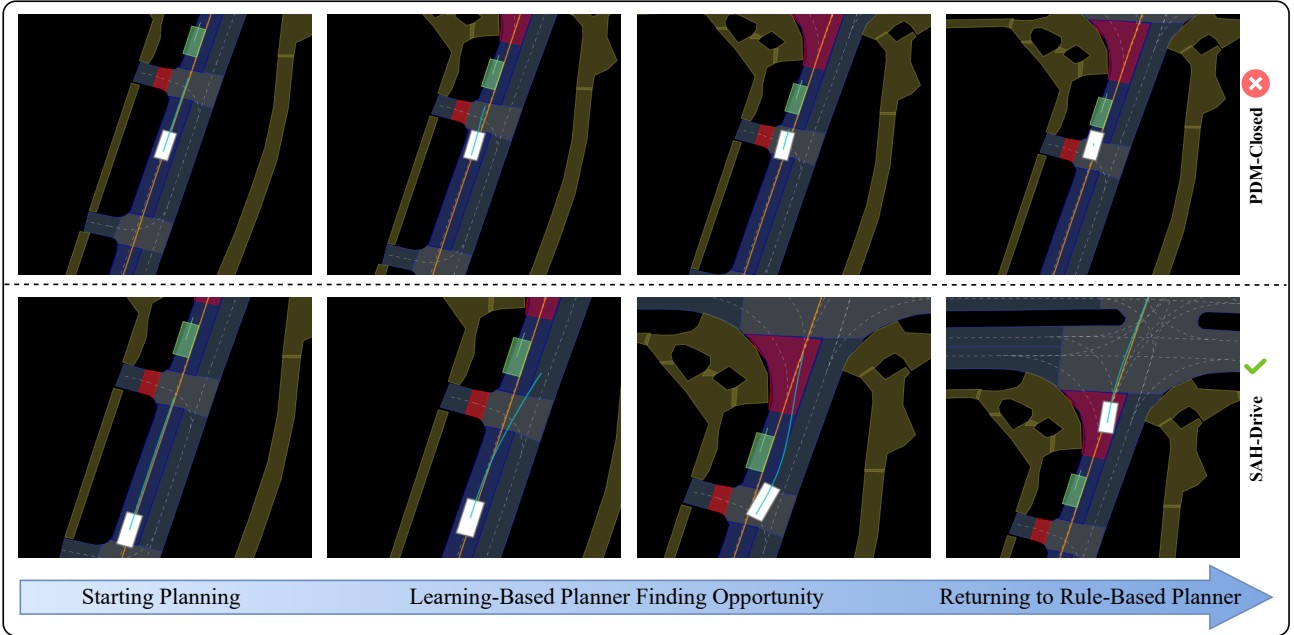

*Figure 5:* **Visualization of a typical long-tail overtaking scenario**. The white rectangle represents the ego vehicle, while the green rectangle indicates an illegally parked vehicle on the road. The overtaking maneuver must be completed for the planning to be considered successful.

.

tion planners with LLM. To further validate this conclusion, we analyzed the impact of model parameters and training dataset size on the performance of SAH-Drive. Detailed experiment analysis can be found in Appendix E.

It is worth noting that although STR2 achieves state-of-the-art performance on Val14 by increasing model parameters and training data through scaling laws, its performance on interPlan is only slightly better than PDM-Closed, due to the lack of long-tail scenarios in nuPlan.

### 4.3. Qualitative Results

Figure 5 is the visualization of a typical long-tail overtaking scenario. The ego vehicle initially adopts a conservative behavior and drives within the current lane. When encountering a long-tail scenario, the learning-based planner achieves a higher score, prompting the planning system to switch to the learning-based planner, continuously seeking opportunities to navigate through the long-tail scenario. After passing through the long-tail scenario, due to the system's preference for the rule-based planner in regular scenarios, the ego vehicle returns to the lane and adopts the conservative strategy again. More qualitative results are shown in Appendix G.

### 4.4. Ablation Study

We conducted ablation experiments to analyze the roles of the rule-based planner and the learning-based planner, as well as the effects of the score-based switching rule, the scenario-based switching rule, and the STDP-based decision neuron in the dual-timescale decision neuron.

In the ablation study of the planner, the variant refers to using only the corresponding planner instead of the complete SAH-Drive. As the simulation results in Table 3 show, the dual planner that combines the rule-based and learning-based planners achieves superior performance compared to using either planner alone. Specifically, the rule-based planner underperforms in scenarios requiring opportunity-seeking capabilities, such as overtaking (-88.46%) and accident zones (-100%), while the learning-based planner struggles in safety-critical scenarios, such as those involving jaywalkers (-6.38%). By integrating both planners, the dual planner achieves better results in both jaywalker and accident zone scenarios.

In the ablation experiment of the dual-timescale decision neuron, the variant refers to using only the corresponding rule or neuron for decision switching. We observe that using only the score-based switching rule leads to performance degradation in accident and construction zone scenarios. Using only the scenario-based switching rule results in a performance drop, mainly in accident and overtaking sce-

narios. When relying solely on the STDP-based decision neuron for planner switching, the performance decline is less pronounced compared to using only the score-based or scenario-based switching rule. The performance deterioration is mainly observed in the construction and overtaking scenarios.

## 4.5. Runtime Analysis

We conducted a runtime analysis of PDM-Closed and SAH-Drive on a computer with an i7-14700KF CPU and an RTX 4080S GPU. The computation time per frame for PDM-Closed is approximately 0.1 seconds, whereas for SAH-Drive, the maximum is around 1 second. In contrast, Diffusion-ES requires about 5 seconds to plan a single trajectory. For large language models, computation time varies depending on the specific model employed. Typically, larger models generate results more slowly and are challenging to apply in real-time scenarios. The computation time for SAH-Drive is acceptable compared to the increased computation time associated with using large language models or other larger models as decision planners.

Figure 6 is the visualization of the runtime analysis. The upper part of the figure corresponds to a regular straight-through intersection scenario from nuPlan, with a duration of 15 seconds, sampled at 0.1-second intervals, resulting in 150 frames (labeled by frame number in the figure). The lower part of the figure corresponds to an overtaking scenario within the interPlan simulation, with a duration of 30 seconds, also sampled at 0.1-second intervals, resulting in 300 frames. The Y-axis in the figure represents the execution time per frame for SAH-Drive and PDM-Closed.

In the regular scenario, the total computation times of PDM-Closed and SAH-Drive were 13.66 seconds and 16.15 seconds, respectively. In the long-tail scenario, the times were 26.18 seconds and 25.17 seconds, respectively. In both scenarios, the two methods exhibited comparable overall computation times. This is due to the use of the diffusion proposal number regulator and the SAH paradigm, which means that longer planning times, such as 1 second, account for only a small portion of the overall SAH-Drive planning process.

## 4.6. Scenario-Aware Characteristics Analysis

Figure 6 also illustrates the switching process between the rule-based planner and the learning-based planner. In the regular scenario, the learning-based planner was active for only 6.70% of the time. In contrast, its activation rate increased to 43.33% in the long-tail scenario, highlighting the scenario-aware nature of our method.

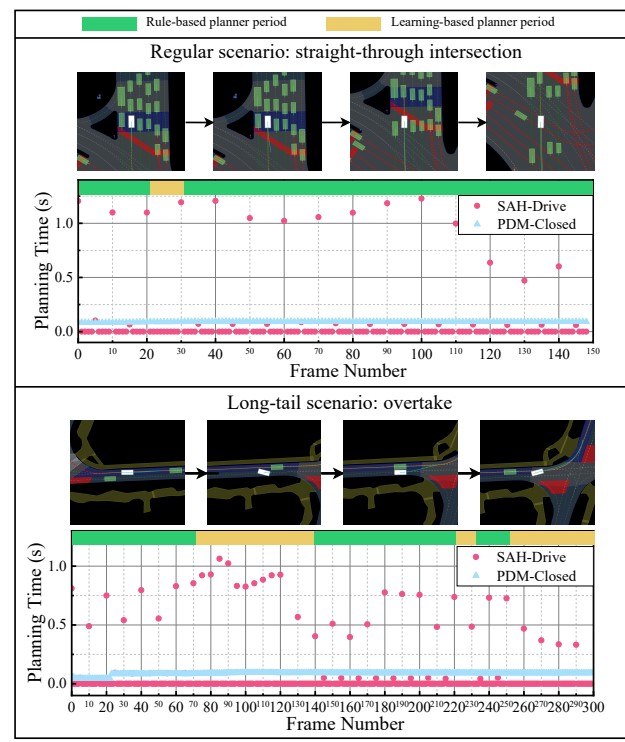

*Figure 6:* **Runtime analysis of SAH-Drive and PDM-Closed under an overtaking scenario**. The red circle represents the planning time of SAH-Drive, while the blue triangle indicates the planning time of PDM-Closed. The green and yellow bars denote the stages where the rule-based and learning-based planners are used, respectively.

## 5. Conclusion

In this study, we proposed SAH-Drive, a planner that extends PDM-Closed by integrating a diffusion model to sample multimodal proposals. SAH-Drive used a scenario-aware hybrid planner paradigm, which employs a dual-timescale decision neuron, incorporating the score-based switching rule, the scenario-based switching rule, and the STDP-based decision neuron to enable flexible transition between the rule-based planner and the learning-based planner. Simulations demonstrate that the proposed SAH-Drive effectively combines the advantages of rule-based and learning-based planners, improving the generalization of the planning system without incurring substantial additional computation time.

**Limitations.** The trajectories sampled by the diffusion model do not fully conform to real-world physical dynamics and can only be regarded as proposals. Future research should focus on harnessing generative models to directly generate trajectories that adhere to real-world physical constraints for autonomous driving.

## Acknowledgements

This work is partially supported by the National Natural Science Foundation of China (No. 52441202), the National Key R&D Program of China (No. 2024YFB43000303, 2023YFB4301802-02), and the Ganwei Program of Beihang University (No. WZ2024-2-16).

## Impact Statement

This paper presents work whose goal is to advance the field of Machine Learning. There are many potential societal consequences of our work, none which we feel must be specifically highlighted here.

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

# A. Diffusion Probabilistic Model for Vehicle Trajectory Generation

## A.1. Diffusion Probabilistic Model

Diffusion probabilistic models are a class of generative models that capture the data distribution by simulating a forward diffusion process, which gradually adds noise to the data, and then learning to reverse this process through denoising. This method, introduced by Sohl-Dickstein et al.(Sohl-Dickstein et al., 2015) and later refined by Ho et al.(Ho et al., 2020), represents the data generation process as a series of steps that involve the iterative removal of noise from a corrupted version of the data.

The forward diffusion process adds Gaussian noise according to a variance schedule $\beta_t$, where the data at timestep $t$, $x_t$, is corrupted as:

$$x_t = \sqrt{\bar{\alpha}_t}x + \sqrt{1 - \bar{\alpha}_t}\epsilon \tag{6}$$

where $\epsilon \sim \mathcal{N}(0, I)$ is Gaussian noise, $\bar{\alpha}_t = \prod_{i=1}^{t} \alpha_i$, $\alpha_i = 1 - \beta_i$ and $\beta_i \in (0, 1)$ is the predefined variance schedule.

The reverse process, denoising, is parameterized by a neural network and can be described by:

$$p_\theta(\tau_0) = \int p(\tau_N) \prod_{i=1}^{N} p_\theta(\tau_{i-1}|\tau_i)d\tau_{1:N} \tag{7}$$

where $\tau_0$ is the original data and $p(\tau_N)$ is a Gaussian prior. The $p_\theta(\tau_{i-1}|\tau_i)$ can be expressed as follows:

$$p_\theta(\tau_{t-1}|\tau_t) = \mathcal{N}(\tau_{t-1}; \mu_\theta(\tau_t, t), \Sigma_\theta(\tau_t, t)) \tag{8}$$

As long as the forward process follows a normal distribution with a sufficiently small variance, $\mu_\theta(\tau_t, t)$ and $\Sigma_\theta(\tau_t, t)$ can be written as follows:

$$\mu_\theta(\tau_t, t) = \frac{1}{\sqrt{\alpha_t}}(\tau_t - \frac{1 - \alpha_t}{\sqrt{1 - \bar{\alpha}_t}}\epsilon_\theta(\tau_t, t))$$
$$\Sigma_\theta(\tau_t, t) = \Sigma(\tau_t, t) = \frac{1 - \bar{\alpha}_{t-1}}{1 - \bar{\alpha}_t}\beta_t \tag{9}$$

By minimizing the variational lower bound on the log-likelihood $\theta^* = \arg\min_\theta -\mathbb{E}_{\tau_0}[\log p_\theta(\tau_0)]$, we can train a neural network $\epsilon_\theta(x_t, t)$ that predicts the added noise.

Furthermore, the framework can be extended by conditioning the reverse process on additional information $c$, such as labels or context. The reverse process becomes:

$$p_\theta(\tau_{t-1}|\tau_t, c) = \mathcal{N}(\tau_{t-1}; \mu_\theta(\tau_t, t, c), \Sigma(\tau_t, t)) \tag{10}$$

The denoising network $\epsilon_\theta(\tau_t, t, c)$ predicts the noise conditioned on $c$, and the loss function is:

$$L(\theta) := \mathbb{E}_{t,\tau_0 \sim q, \epsilon \sim \mathcal{N}}\left[\|\epsilon - \epsilon_\theta(\tau_t, t, c)\|^2\right] \tag{11}$$

Following previous work(Janner et al., 2022; Yang et al., 2024), in order to use the diffusion model for autonomous vehicle trajectory generation, we represent the trajectory of the vehicle as follows and use it as the input to the diffusion model.

$$\tau = \begin{bmatrix} x_0 & x_1 & ... & x_T \\ y_0 & y_1 & ... & y_T \\ \theta_0 & \theta_1 & ... & \theta_T \end{bmatrix} \tag{12}$$

Where $x$ and $y$ represent the longitudinal and lateral offsets, respectively, and $\theta$ is the heading angle.

### A.2. Closed-loop Trajectory Planning

Closed-loop trajectory planning is a crucial component in the autonomous driving pipeline, ensuring that a vehicle's path is dynamically adjusted in real-time based on its current state, environmental conditions, and unforeseen disturbances.

Mathematically, closed-loop trajectory planning can be formulated as an optimization problem, where the objective is to minimize a cost function while considering constraints such as vehicle dynamics, safety margins, and environmental obstacles. The trajectory $\mathbf{x}(t) = [x(t), y(t), \theta(t)]^T$ represents the state of the vehicle at time $t$. The control inputs $\mathbf{u}(t) = [v(t), \delta(t)]^T$ include the vehicle's speed $v(t)$ and steering angle $\delta(t)$. The dynamic model governing the vehicle's motion, typically described by a bicycle model or more advanced kinematic models, can be written as:

$$
\begin{bmatrix} \dot{x}(t) \\ \dot{y}(t) \\ \dot{\theta}(t) \end{bmatrix} = \mathbf{f}(\mathbf{x}(t), \mathbf{u}(t)) \tag{13}
$$

where $\mathbf{f}$ is a nonlinear function describing the vehicle's motion. In this paper, the dynamic model used is the **bicycle model**, and the objective function is the **PDM score**. The PDM Score is a metric used to evaluate driving agent trajectories, computed by aggregating subscores into a value in the range [0, 1], and is an efficient reimplementation of the nuPlan closed-loop score metric.

## B. The Detailed Design of the Diffusion Proposal Generation Model

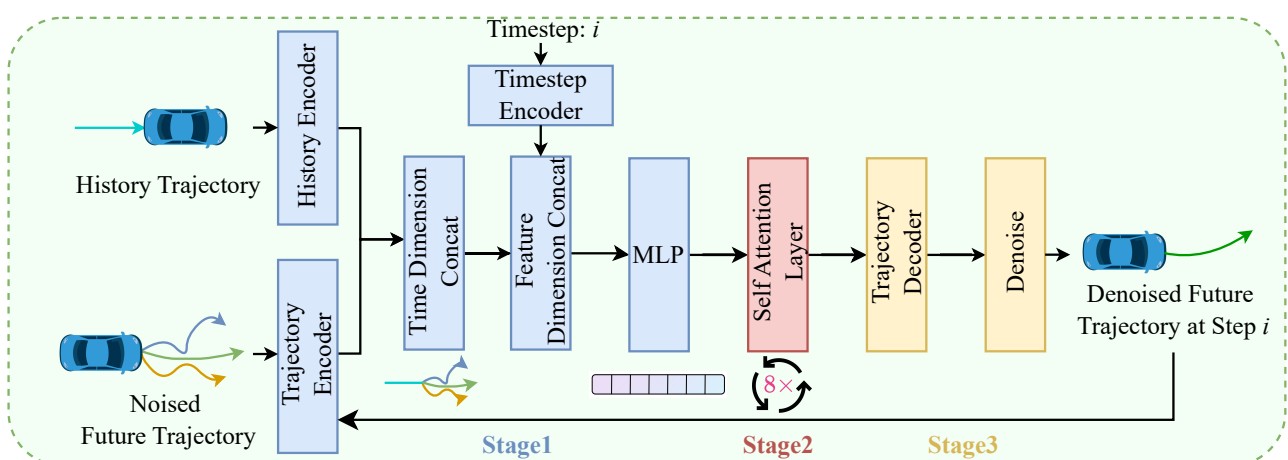

*Figure 7:* The overall architecture of the diffusion proposal generation model

**1. Feature Fusion**: This stage integrates inputs to construct a unified feature representation for proposal generation. First, the historical trajectory and noised future trajectory are encoded into a history embedding and trajectory embedding, respectively. Let the historical trajectory be represented as $\mathbf{x}_h = [\mathbf{x}_{h,1}, \mathbf{x}_{h,2}, \ldots, \mathbf{x}_{h,T}]$, and the noisy trajectory as $\mathbf{x}_n = [\mathbf{x}_{n,1}, \mathbf{x}_{n,2}, \ldots, \mathbf{x}_{n,T}]$, where $\mathbf{x}_{h,t}, \mathbf{x}_{n,t} \in \mathbb{R}^3$ denote the positional information at the $t$-th time step of the respective trajectories. We then compute the first-order and second-order differences of $\mathbf{x}_h$ and $\mathbf{x}_n$, corresponding to velocity and acceleration, and denote them as $\mathbf{v}_h$, $\mathbf{v}_n$, $\mathbf{a}_h$, and $\mathbf{a}_n$, respectively. The position, velocity, and acceleration information of the trajectories are concatenated to form feature vectors $\mathbf{F}_h = [\mathbf{x}_h, \mathbf{v}_h, \mathbf{a}_h]$ and $\mathbf{F}_n = [\mathbf{x}_n, \mathbf{v}_n, \mathbf{a}_n]$. These feature vectors are then mapped to a fused representation $\mathbf{h}_h$ and $\mathbf{h}_n$ through a Multi-Layer Perceptron (MLP). Then, $\mathbf{h}_h$ and $\mathbf{h}_n$ are concatenated along the temporal dimension and further combined with the timestep embedding along the feature dimension. The information from the timestep is then integrated through an additional MLP.

**2. Self-Attention Fusion**: The feature embedding is refined using self-attention to model global spatial and temporal dependencies. Time position embeddings are added to provide context, and the refined representation passes through eight layers of self-attention. This process enhances contextual understanding and ensures that both spatial relationships and temporal coherence are captured effectively.

**3. Decoding and Denoising**: The final stage reconstructs the trajectory through iterative refinement. A trajectory decoder predicts trajectory noise, which is progressively used to correct the noised future trajectory by a denoising module over multiple diffusion steps. Although the diffusion model can effectively capture physical information, the generated trajectories do not strictly adhere to physical constraints and serve merely as proposals that require further refinement through simulation to ensure physical feasibility.

## C. Planning Baselines

We reviewed recent outstanding baselines for closed-loop trajectory planning, as outlined below:

1. **PDM-Closed** (Dauner et al., 2023): a rule-based planner that borrows the concept of model predictive control (MPC), using forecasting, proposals, simulation, scoring, and selection to get the trajectory with the highest score.
2. **Diffusion-ES** (Yang et al., 2024): a learning-based planner that combines a diffusion model and evolutionary search, iteratively evolves to obtain the best trajectory.
3. **STR2** (Sun et al., 2024): a scalable, MoE-based autoregressive motion planner that leverages ViT and causal transformers to achieve generalization and scalability on diverse urban driving scenarios.
4. **IDM** (Treiber et al., 2000): a car-following model designed for safe and realistic traffic flow simulations, emphasizing accident prevention and maintaining a safe distance to the leading vehicle by adjusting its speed.
5. **Urban Driver** (Scheel et al., 2022): a policy gradient method leveraging a differentiable simulator and mid-level representations to efficiently learn and generalize imitative driving policies for complex urban scenarios from large-scale real-world data.
6. **Game Former** (Huang et al., 2023): a learning-based planner that employs hierarchical game theory and a transformer-based architecture to model interactive behaviors between traffic participants.
7. **DTPP** (Huang et al., 2024): a differentiable joint training framework that integrates ego-conditioned motion prediction and learnable context-aware cost evaluation within a tree-structured policy planner.
8. **HybridLLMPlanner** (Hallgarten et al., 2024): a two-stage hybrid planner that combines LLM with PDM-Closed, where LLM is used for behavior planning and the PDM-Closed is used for motion planning.
9. **Diffusion Planner** (Zheng et al., 2025): Utilizes a transformer-based diffusion model to produce trajectories without rule-based heuristics by jointly handling prediction and planning, guided by a classifier for high-quality sampling.
10. **PlanTF** (Cheng et al., 2024b): An imitation-based planner that focuses on essential ego features and effective data augmentations to reduce compounding errors and mitigate the imitation gap.
11. **Pluto** (Cheng et al., 2024a): An imitation learning planner featuring a longitudinal-lateral aware architecture, contrastive learning, and efficient auxiliary loss.

*Table 4:* **Closed-loop metric of the PDM score**. It consists of multiplicative and weighted metrics, where curly brackets denote discrete value sets, and square brackets represent continuous ranges.

| Type | Metric | Weight | Range |
|---|---|---|---|
| Multiplicative | No at-fault Collisions (NC) | Multiplier | $\{0, \frac{1}{2}, 1\}$ |
| Multiplicative | Drivable Area Compliance (DAC) | Multiplier | $\{0, 1\}$ |
| Multiplicative | Driving Direction Compliance (DDC) | Multiplier | $\{0, \frac{1}{2}, 1\}$ |
| Multiplicative | Making Progress (MP) | Multiplier | $\{0, 1\}$ |
| Weighted | Time to Collision (TTC) | 5 | $\{0, 1\}$ |
| Weighted | Ego Progress (EP) | 4 | $[0, 1]$ |
| Weighted | Speed-limit Compliance (SC) | 1 | $[0, 1]$ |
| Weighted | Comfort (C) | 1 | $\{0, 1\}$ |

## D. PDM Score

The closed-loop score consists of weighted metrics and multiplier metrics, assessing the safety, compliance, and driving quality of an autonomous driving system, which is shown in Table 4. The weighted metrics include Time to Collision (TTC), Ego Progress (EP), Speed-limit Compliance (SC), and Comfort (C), which evaluate collision risk, driving efficiency, speed regulation, and driving comfort. The multiplier metrics—No at-fault Collisions (NC), Drivable Area Compliance (DAC),

Driving Direction Compliance (DDC), and Making Progress (MP)—penalize violations such as at-fault collisions, driving outside allowed areas, wrong-way driving, and failure to maintain reasonable progress. The scoring framework integrates these factors to comprehensively assess the planner's performance in autonomous driving tasks.

## E. The Impact of Model Parameter Size and Training Dataset Scale

To further validate that the dual planner paradigm reduces data requirements and enables simpler structures, we scaled the feature dimensions and adjusted the training dataset composition. The results of the impact of model parameter size are shown in Table 5. The STR2 model has 800M parameters, with the training dataset being the entire nuPlan training dataset. In comparison, SAH-Drive significantly reduces both model size and training dataset, while performing better in interPlan. The interPlan score is highest when the feature dimension is 16. Both the larger and smaller feature dimensions lead to a decrease in the interPlan score.

*Table 5:* **The impact of model parameter size**. The original SAH-Drive has a feature dimension of 16, and its model parameter count is only 0.0355% of STR2.

|  | Feature Dimension | Model Parameters | Model Occupancy | interPlan Score |
|---|---|---|---|---|
|  | 8 | 143k | 0.55Mb | 53 |
| SAH-Drive | 16 | 284k | 1.08Mb | 64 |
|  | 32 | 581k | 2.22Mb | 56 |
| STR2 | 512 | 800m | / | 46 |

*Table 6:* **The impact of training dataset size**. The dataset sizes for nuPlan_mini, Singapore, and Boston are 7.96 GB, 32.56 GB, and 35.54 GB, respectively. The score of SAH-Drive on interPlan decreases as the dataset size increases.

| Dataset Description | Dataset Size | interPlan Score |
|---|---|---|
| nuPlan Mini | 7.96Gb | 65 |
| nuPlan Mini+Singapore | 40.52Gb | 61 |
| nuPlan Mini+Singapore+Boston | 76.06Gb | 59 |

As shown in Table 6, the planner's score decreases when trained on a larger dataset. This suggests a potential mismatch between the model capacity and the complexity of the expanded data, or that the additional data introduces distributional challenges. This observation highlights that, within the scenario-aware hybrid planner paradigm, the learning-based planner can be deliberately kept lightweight and trained on targeted long-tail scenarios, leaving the rule-based planner to handle more regular cases. Nonetheless, designing a more complex learning-based planner using scaling laws and training it on large-scale datasets remains a promising alternative.

## F. Further Validation on the Effectiveness of the SAH Paradigm

*Table 7:* **Experimental results of PDM+PlanTF and PDM+Pluto combined under the SAH paradigm**. The highest score is indicated in **bold**, and the percentage increase in scores of PlanTF and Pluto after combining PDM under the SAH paradigm, compared to their original versions, is underlined.

| Planner | interPlan | Val14 (R) | Val14 (NR) | Test14-Random (R) | Test14-Random (NR) | Test14-Hard (R) | Test14-Hard (NR) |
|---|---|---|---|---|---|---|---|
| Pluto | 48 | 78 | **89** | 78 | **89** | 60 | 70 |
| SAH (PDM+Pluto) | 49(2%↑) | 79(1%↑) | **89** | 88(13%↑) | 88 | 81(35%↑) | **78**(11%↑) |
| PlanTF | 33 | 77 | 84 | 80 | 85 | 61 | 69 |
| SAH (PDM+PlanTF) | 40(21%↑) | 78(1%↑) | 86(2%↑) | 86(8%↑) | 87(2%↑) | 80(31%↑) | 77(12%↑) |
| SAH-Drive (Ours) | **64** | **90** | **89** | 87 | 86 | **83** | **78** |

We use PlanTF and Pluto as learning-based planners and PDM-Closed as the rule-based planner, combining them within the SAH paradigm for simulation experiments. The results are summarized in Table 7. As shown in the table, both PlanTF

and Pluto achieve consistent improvements across various nuPlan splits when integrated with PDM-Closed under the SAH paradigm, outperforming their original versions. The performance gains are particularly notable on Test14-Hard (R), with improvements of 35% and 31%, respectively.

## G. More Qualitative Results

As shown in Figure 8, SAH-Drive successfully plans in construction zones, jaywalker, nudge, and accident scenarios, whereas PDM-Closed fails in both the construction zone and accident scenarios, as it stops behind the obstructing vehicle and is unable to change lanes. This demonstrates that SAH-Drive has better generalization ability than PDM-Closed and is more capable of handling planning in long-tail scenarios. The left side of the figure shows the scenario corresponding to the visualized simulation results in the middle, while the right side indicates the planner used and whether the planning was successful.

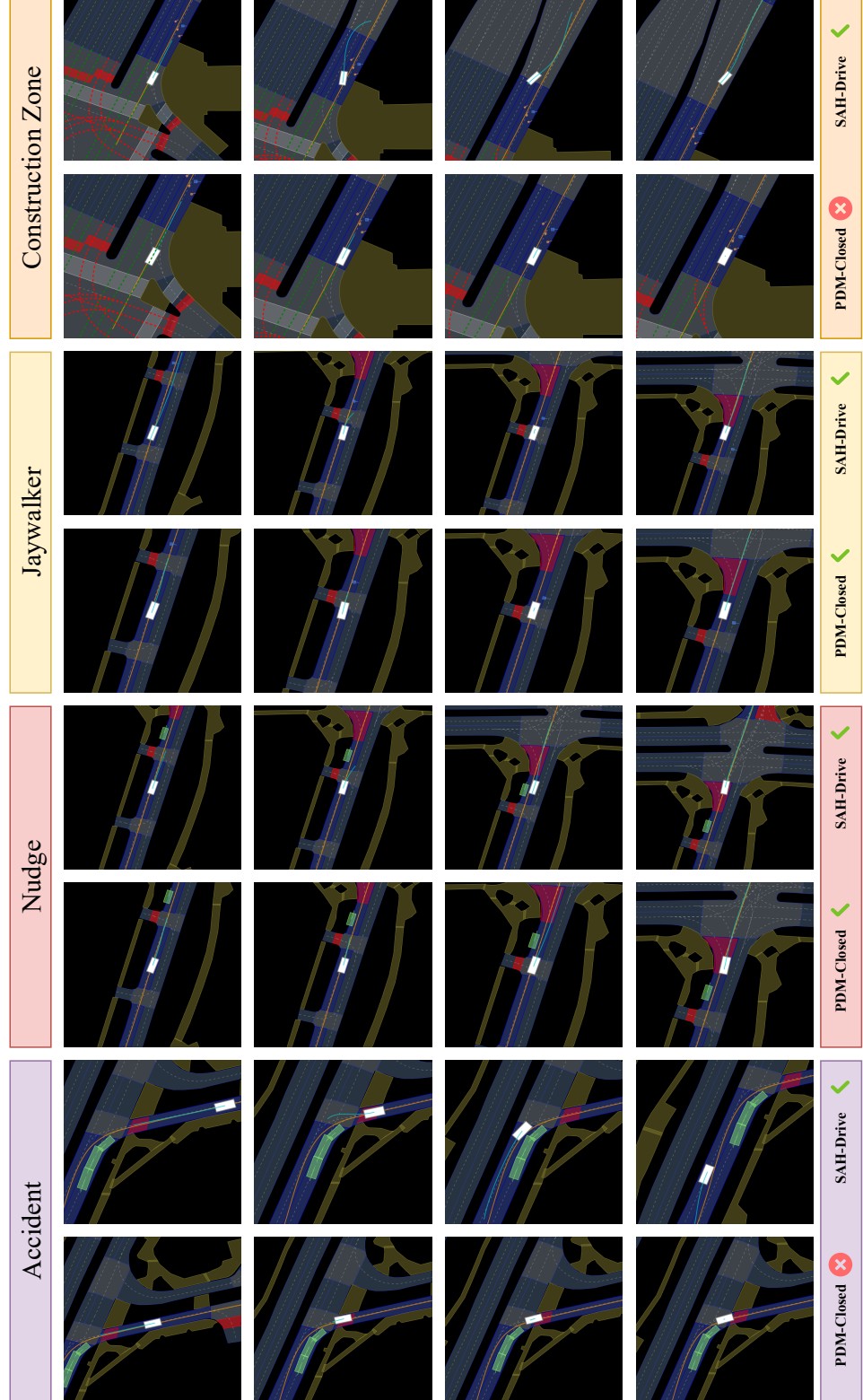

*Figure 8:* More qualitative results of SAH-Drive and PDM-Closed

