# OpenReview forum: "SAH-Drive: A Scenario-Aware Hybrid Planner for Closed-Loop Vehicle Trajectory Generation"
_ICML.cc/2025/Conference — ICML 2025 poster_

### Official Review · Reviewer_2ndh · 2025-02-25

**Overall Recommendation:** 4

**Summary:**

This paper presents a hybrid planner, SAH-Drive, for closed-loop vehicle trajectory generation.

SAH-Drive uses the fast-slow hybrid planner paradigm. It has a rule-based fast planning path using PDM and a learned slow planning path using a diffusion model. The main innovation of SAH-Drive is that it uses a scenario-based switching module that switches between the two planners based on the scenario.

The authors evaluated the SAH-Drive hybrid planner on the standard nuPlan dataset as well as the new interPlan dataset. The result shows that SAH-Drive achieves better scores than the previous state-of-the-art planners.

Because of the hybrid approach, the authors also show that SAH-Drive has a much faster runtime than a purely-learned planner.


## update after rebuttal

No updates.

**Claims And Evidence:**

* The claims are relatively well-supported by the evaluation results.
* However, even though SAH-Drive has a scenario-aware switching logic, which is different from the existing hybrid planner approaches, the novelty is not strong.

**Essential References Not Discussed:**

N/A

**Experimental Designs Or Analyses:**

* Experimental design is solid.

**Methods And Evaluation Criteria:**

* The evaluation is relatively thorough. The authors evaluated their SAH-Drive approach on both the standard nuPlan validation set and the new interPlan dataset. The authors also compared the runtime speed of different planner approaches.

**Other Comments Or Suggestions:**

N/A

**Other Strengths And Weaknesses:**

N/A

**Questions For Authors:**

N/A

**Relation To Broader Scientific Literature:**

N/A

**Theoretical Claims:**

* There are no theoretical proofs in the paper.

---

> ### Author Rebuttal · Authors · 2025-04-01
>
> We really appreciate the reviewer’s constructive comments and positive feedback, which have helped us better articulate our contributions and clarify the novelty of our approach. Regarding the concerns of the reviewer 2ndh, we provide the following responses.
>
> > Q1: Even though SAH-Drive has a scenario-aware switching logic, which is different from the existing hybrid planner approaches, the novelty is not strong.
>
> Thanks for your valuable comments. The novelty of this work lies not only in proposing the **Scenario-Aware Hybrid Planner Paradigm** but also in the following key aspects:
>
> - **STDP-Based Decision Neuron**: It memorizes the performance of both planners during the planning process, enabling adaptive switching based on scenario characteristics.
> - **Proposal Number Regulator**: It dynamically adjusts the number of diffusion model samples in real-time, improving planning efficiency without sacrificing trajectory diversity.
> - **Trajectory Fusion Mechanism**: We propose an exponential-weighted fusion of the highest-scoring diffusion and PDM trajectories to balance trajectory diversity and driving safety.
> - **Efficient Learning-Based Planner Adaptation**: SAH-Drive reduces the required training data compared to fully learning-based planners (from **285.61GB to 8.25GB**, a reduction of approximately **97%**) while maintaining planning performance, especially in long-tail scenarios such as interPlan and Test14-hard (see rebuttal table 2 https://anonymous.4open.science/r/SAH-Drive-Pic-9E34/benchmark_table.png for details). This makes targeted training on rare, real-world long-tail scenarios both feasible and efficient.
>
> We believe that **adapting learning-based planners to long-tail scenarios with limited data** is a promising direction for future research, and our work lays the foundation for this advancement. We will further highlight the novelty in the final version.
>
> Once again, we sincerely appreciate your valuable insights, and we hope this clarification strengthens the perceived novelty and contributions of our work.

---

> > ### Comment · Reviewer_2ndh · 2025-04-04
> >
> > Thank you for your response.

---

> > > ### Author Response · Authors · 2025-04-07
> > >
> > > Thank you for the response and recognition again! We truly appreciate your valuable review for improving our work.

---

### Official Review · Reviewer_Z9oQ · 2025-03-09

**Overall Recommendation:** 3

**Summary:**

This work introduces a hybrid planning framework that combines diffusion-based and rule-based proposal generation, evaluated through PDM simulation, with asynchronous updates and spiking neurons guiding final trajectory selection. To optimize efficiency and performance, an adaptive proposal regulator dynamically adjusts the number of diffusion proposals based on overall scoring. Additionally, a STDP-based decision neuron enables adaptive switching between rule-based and learning-based proposals by considering time delays and long-short-term synaptic plasticity rules. The proposed planner achieves compelling results on the InterPlan long-tail benchmark in nuPlan.

**Claims And Evidence:**

The writing is generally clear, with some minor ambiguities in the methodology. The section on the proposal regulator would benefit from a more detailed explanation, preferably formulated with equations to clarify its probabilistic mechanism. Additionally, as a non-expert in spiking networks, I recommend including an algorithm to explicitly illustrate the switching workflow, improving overall readability.

**Essential References Not Discussed:**

[1] Cheng, J., Chen, Y., & Chen, Q. (2024). Pluto: Pushing the limit of imitation learning-based planning for autonomous driving. arXiv preprint arXiv:2404.14327.

[2] J. Cheng, Y. Chen, X. Mei, B. Yang, B. Li, and M. Liu, “Rethinking imitation-based planner for autonomous driving,” in International Conference on Robotics and Automation (ICRA), 2024.

[3] Zheng, Y., Liang, R., Zheng, K., Zheng, J., Mao, L., Li, J., ... & Liu, J. (2025). Diffusion-Based Planning for Autonomous Driving with Flexible Guidance. arXiv preprint arXiv:2501.15564.

[4] Liu, H., Chen, L., Qiao, Y., Lv, C., & Li, H. (2024). Reasoning Multi-Agent Behavioral Topology for Interactive Autonomous Driving. arXiv preprint arXiv:2409.18031.

[5] Zhang, D., Liang, J., Guo, K., Lu, S., Wang, Q., Xiong, R., ... & Wang, Y. (2025). CarPlanner: Consistent Auto-regressive Trajectory Planning for Large-scale Reinforcement Learning in Autonomous Driving. arXiv preprint arXiv:2502.19908.

**Experimental Designs Or Analyses:**

Another key issue is the lack of comparison with recent state-of-the-art methods on the benchmark. Specifically, results should be compared against pure imitation learning (IL)-based planners [1,2] and diffusion-based planners [3] to better contextualize the proposed approach. Additionally, the metric selection is incomplete, as it does not report the non-reactive closed-loop score (CLS-NR), which is crucial for evaluating the closed-loop capability of logged behavior. The ablation study also lacks generality—given that this work presents a general hybrid paradigm, further experimental validation is needed by integrating PDM with different types of planners [1-3] to strengthen the generalizability of the findings. Also, it is recommended to provide extra results under different time variance between rule and learning-based planner for further justifications.

Reference:

[1] Cheng, J., Chen, Y., & Chen, Q. (2024). Pluto: Pushing the limit of imitation learning-based planning for autonomous driving. arXiv preprint arXiv:2404.14327.

[2]J. Cheng, Y. Chen, X. Mei, B. Yang, B. Li, and M. Liu, “Rethinking imitation-based planner for autonomous driving,” in International Conference on Robotics and Automation (ICRA), 2024.

[3] Zheng, Y., Liang, R., Zheng, K., Zheng, J., Mao, L., Li, J., ... & Liu, J. (2025). Diffusion-Based Planning for Autonomous Driving with Flexible Guidance. arXiv preprint arXiv:2501.15564.

**Methods And Evaluation Criteria:**

The benchmark selection is suitable but lacks further comparison. For instance, it would be valuable to analyze the performance on Test14-hard, which evaluates scenarios where purely rule-based methods struggle. A detailed comparison here would better highlight the necessity of integrating a learning-based planner for enhanced support and robustness.

**Other Comments Or Suggestions:**

If the authors could provide additional compelling results to demonstrate the generality of this approach, such as applying SAH+[1,2], and extend evaluations across additional benchmarks (Test14-hard), I would consider raising my score.

**Other Strengths And Weaknesses:**

N/A.

**Questions For Authors:**

Refer to Experimental Designs Or Analyses

**Relation To Broader Scientific Literature:**

This hybrid framework may presents potential in combining further learning-based planners.

**Theoretical Claims:**

No theoretical claims.

---

> ### Author Rebuttal · Authors · 2025-04-01
>
> We sincerely appreciate the reviewer’s constructive feedback. In response to the concerns raised by Reviewer Z9oQ, we provide the following responses.
>
> > Q1: Detailed explanation on the proposal regulator.
>
> The number of diffusion trajectories $N'$ is dynamically adjusted based on the highest diffusion trajectory score $S_{\max}$ relative to a threshold $\tau$.
>
> The trajectory count is updated based on the highest trajectory score:
> $$N' =
>    \begin{cases}
>    \frac{N}{2}, & S_{\max} > \tau, \\
>    2N, & S_{\max} < \tau.
>    \end{cases}$$
> To regulate trajectory count, we apply:
> $$N' = \max(N_{\min}, \min(N', N_{\max})).$$
> We will add the detailed explanation of the proposal number regulator in the final version.
>
> > Q2: Including an algorithm to explicitly illustrate the switching workflow
>
> We outline the decision process (Fig. 3, p5 l220 in the original paper) and will add a detailed algorithm in the revised paper.
>
> **Input**: Rule-based planner score $s_c$, Learning-based planner score $s_r$
> **Output**: Selected planner
> 1. Update $w_r$ and $w_c$ using Equation (3) from the paper.
> 2. Update consecutive counts $n_e$​ and $n_p$.
> 3. assign the decision category: category $\leftarrow$ decision_space$(s_r, s_c,n_e,n_p)$
> 4. Select the planner based on the decision category:
> 	- If category = score, then planner $\leftarrow$ score_rule$(s_r, s_c)$
> 	- If category = scenario, then planner $\leftarrow$ scenario_rule$(n_e, n_p)$
> 	- Otherwise, planner $\leftarrow$ STDP_neuron$(w_r, w_c)$
>
> > Q3: Lack of comparison with recent SOTA methods and further comparison on Test14-hard.
>
> We have extended our benchmark evaluation to include **recent learning-based SOTA methods**: Pluto, PlanTF, and DiffusionPlanner. The results are summarized in the table below:
>
> | Planner           | interPlan | Val14 (R) | Val14 (NR) | test14 (R)   | test14 (NR) | test14-hard (R) | test14-hard (NR) |
> | ----------------- | --------- | --------- | ---------- | ------------ | ----------- | --------------- | ---------------- |
> | Pluto             | 48        | 78        | 89         | 78           | **89**      | 60              | 70               |
> | SAH (PDM+Pluto)   | 49(2%↑)   | 79(1%↑)   | 89         | **88**(13%↑) | 88          | 81(35%↑)        | **78**(11%↑)     |
> | PlanTF            | 33        | 77        | 84         | 80           | 85          | 61              | 69               |
> | SAH (PDM+PlanTF)  | 40(21%↑)  | 78(1%↑)   | 86(2%↑)    | 86(8%↑)      | 87(2%↑)     | 80(31%↑)        | 77(12%↑)         |
> | Diffusion Planner | 24        | 83        | **90**     | 83           | **89**      | 69              | 75               |
> | SAH-Drive         | **64**    | **90**    | 89         | 87           | 86          | **83**          | **78**           |
>
> As shown in the table, SAH-Drive consistently outperforms SOTA learning-based planners across multiple evaluation metrics. Notably, despite being trained only on nuPlan-mini, SAH-Drive achieves **SOTA scores on interPlan and test14-hard**, while also delivering **near-SOTA performance on other benchmarks**. The full experimental results, including all baselines discussed in the paper, can be found at https://anonymous.4open.science/r/SAH-Drive-Pic-9E34/benchmark_table.png.
>
> > Q4: Further experimental validation is needed by integrating PDM with different types of planners such as SAH + PlanTF and Pluto.
>
> Thanks for pointing this out. We employed the SAH + PlanTF and Pluto as the reviewer requested. The results are summarized in the table under Q3. As shown in the table, both PlanTF and Pluto achieve **consistent improvements across various nuPlan splits** when integrated with PDM under the SAH framework, outperforming their original versions. The performance gains are particularly notable on Test14-hard (R), with improvements of **35% and 31%**, respectively.
>
> > Q5: Extra results under different time variance between rule and learning-based planner for further justifications.
>
> To complement our analysis of time variance between rule-based and learning-based planners, we conducted an additional experiment using SAH-Drive in the regular straight-through intersection nuPlan scenario. The visualization is available at https://anonymous.4open.science/r/SAH-Drive-Pic-9E34/time_variance_visualization.png. The learning-based planner (yellow) was active **6.70%** overall. In contrast, its activation rate **increased** to **43.33%** in the long-tail scenario, highlighting the scenario-aware nature of our method. We will include the above details in the revised paper.
>
> > Q6: Insufficient references
>
> Thank you for your suggestion. We will incorporate **Pluto, PlanTF, Diffusion Planner, CarPlanner, and BeTop** into the related work section to provide a more comprehensive discussion of prior research.

---

> > ### Comment · Reviewer_Z9oQ · 2025-04-02
> >
> > Thank you for the detailed experiments and thorough explanations. I believe most of my concerns have been fully addressed, and I will therefore increase my rating accordingly.

---

> > > ### Author Response · Authors · 2025-04-07
> > >
> > > Thanks for your feedback and raising the score. Your time and effort in reviewing our work are truly appreciated!  We will revise the manuscript according to your valuable comments.

---

### Official Review · Reviewer_NL4n · 2025-03-12

**Overall Recommendation:** 3

**Summary:**

This paper performs post-ensemble on two SOTA methods on the nuPlan Val14 dataset. Drawing inspiration from Spike-Timing Dependent Plasticity（STDP）in neuroscience, it proposes a novel scoring method called Score-based STDP. Additionally, it incorporates both a Scenario-based switching rule and a Score-based switching rule to evaluate the generated trajectories. Experimental results show that the proposed approach achieves performance close to SOTA on the nuPlan Val14 dataset and achieves SOTA on the interPlan dataset.

**Claims And Evidence:**

The authors claim the effectiveness of the proposed method by demonstrating SOTA performance on the interPlan dataset. However, on the nuPlan Val14 dataset, its performance is lower than that of the two ensemble methods it utilizes. The optimal parameters for the PDM scorer vary across different scenarios/sub-dataset of nuPlan, and simply adjusting these parameters can lead to significantly different results. Therefore, achieving SOTA only on the interPlan may be not enough, especially given that it fails to outperform the ensemble methods this paper used on Val14.

**Essential References Not Discussed:**

NA

**Experimental Designs Or Analyses:**

Yes. Fig. 6 is intuitive, allowing readers to understand in which scenarios the planner switching occurs. From the figure, it seems that the switch is triggered by a relatively low Ego progress score of the PDM-closed planner. This effectively addresses the difficulty of lane-changing in rule-based methods.

**Methods And Evaluation Criteria:**

Strength:
Integrating learning-based and rule-based methods to address the long-tail problem and the computational efficiency issues of learning-based approaches is a practical direction.
Weakness:
However, with the adoption of a post-ensemble approach, most aspects aside from STDP feel somewhat unremarkable. The two ensemble methods used belong to others, while the Scenario-based and Score-based switching rules seem somewhat naïve.

**Other Comments Or Suggestions:**

NA

**Other Strengths And Weaknesses:**

Strength：

The use of the diffusion proposal number regulator significantly reduces planning time. SAH achieves planning times even shorter than the rule-based planner PDM-Closed, enhancing the efficiency of learning-based planners in real-world applications.

Weakness：

The paper still relies on the PDM scorer to evaluate planner-generated trajectories, which often requires precise lane markings and surrounding vehicle motion information. Such information is often expensive to obtain in real-world scenarios.

**Questions For Authors:**

NA

**Relation To Broader Scientific Literature:**

The Score-based STDP used in paper is related to the Spike-Timing Dependent Plasticity in neuroscience.

**Theoretical Claims:**

Score-based STDP is interesting. It seems to effectively integrate two different methods in interPlan, but my concern is that it isn’t effective in val14, raising questions about its generalization ability.

---

> ### Author Rebuttal · Authors · 2025-04-01
>
> We thank the reviewer for the constructive comments. Regarding the concerns of the reviewer NL4n, we provide the following responses.
>
> > Q1: Achieving SOTA only on the interPlan may not be enough, especially given that it fails to outperform the ensemble methods this paper used on Val14.
>
> Thanks for the constructive comment. We understand why the reviewer might perceive a **"1+1 < 2" issue**—where integrating PDM-Closed and Diffusion-ES within the SAH paradigm appears to yield a lower score. However, this is a misunderstanding. The learning-based planner used in SAH-Drive is not the original Diffusion-ES, but rather a simplified version that removes evolutionary search and reduces the model complexity.
>
> We simplify the Diffusion-ES to **improve planning efficiency**, achieving a **nearly 98% reduction in average per-frame runtime—from 5s to just 0.1s**. Additionally, our optimization **drastically reduces** the model’s memory footprint (**from 25.62MB to 1.08MB, a 96% decrease**) and parameter count (**from 6.7M to 284K, also a 96% reduction**). This lightweight design enables the model to achieve strong performance even when trained solely on nuPlan-mini. However, this efficiency gain comes with a slight performance trade-off.
>
> As shown in the table below, when evaluated on Val14, this simplified diffusion-based planner score is **88**. Additionally, SAH-Drive’s rule-based planner (PDM-Closed) operates at 5Hz, and its standalone performance on Val14 is **90**. When combining these two planners within the SAH paradigm, the overall performance remains **90**, which does not indicate a drop in performance compared to its individual components. This directly addresses the reviewer’s concern.
>
> | Planner                                                         | interPlan | Val14  |
> | --------------------------------------------------------------- | --------- | ------ |
> | PDM-Closed 5Hz                                                  | 40        | **90** |
> | Simplified diffusion-based planner                              | 62        | **88** |
> | SAH-Drive (PDM-Closed 5Hz + Simplified diffusion-based planner) | 64        | **90** |
>
> > Q2: Score-based STDP is interesting, but my concern is that it isn’t effective in val14, raising questions about its generalization ability.
>
> Thanks for pointing this out. Regarding generalization concerns, we conducted further experiments on two extra datasets: Test14-Random and Test14-hard. Results on **interPlan, Val14, Test14-random, and Test14-hard** are shown in rebuttal table 2 (https://anonymous.4open.science/r/SAH-Drive-Pic-9E34/benchmark_table.png). SAH-Drive achieves SOTA on interPlan and Test14-hard and near-SOTA on Val14 and Test14-random, which confirms SAH-Drive’s robustness and generalizability across diverse scenarios.
>
> > Q3: Most aspects aside from STDP feel somewhat unremarkable and the Scenario-based and Score-based switching rules seem somewhat naive.
>
> We appreciate the reviewer for the comment but disagree with our highest respect that most aspects aside from STDP are unremarkable. The Scenario-based and Score-based switching rules play a crucial role in complementing the STDP-based decision neuron.
>
> The Scenario-based and Score-based rules provide immediate decision-making capabilities, analogous to Short-Term Plasticity in biological neurons, which enables rapid adaptation to sudden environmental changes. If we were to rely solely on the STDP-based decision neuron, its memory-driven nature—which accounts for historical planning performance—could lead to delayed responses to sudden, strong signals. This delay in planner switching could be problematic, especially in autonomous driving scenarios where immediate adaptation is critical. The ablation experiment table in the original paper (p7, l350) further validates this, showing that without switching rules, the planner’s interPlan score decreases by **3.13%**. Unlike approaches that emphasize model complexity and large-scale datasets, SAH-Drive reduces reliance on extensive training data through flexible planner switching.
>
>
> > Q4: The paper still relies on the PDM scorer to evaluate planner-generated trajectories, which often requires precise lane markings and surrounding vehicle motion information. Such information is often expensive to obtain in real-world scenarios.
>
> Thanks for your valuable comments, and we understand your concerns. However, our study primarily focuses on **improving the trajectory planning paradigm and the quality of its planned trajectories** given well-perceived environmental information, rather than addressing the perception challenge itself. In real-world scenarios, high-fidelity perception systems in autonomous vehicles already provide such environmental information, **ensuring that our approach remains applicable in practical deployments**.

---

### Decision · Program_Chairs · 2025-05-01

**Decision:**

Accept (poster)

**Comment:**

The paper proposes a scenario-aware hybrid planner for closed-loop trajectory prediction. It combines a lightweight rule-based planner and a comprehensive learning-based planner. The motivation is clearly stated with comprehensive experiments.

All reviewers (two short reviews) reach concensus that the manuscript is in good shape. AC step in, read the paper and rebuttal. The methodology seems to be naive and simple (raised by reviewers as well). Given the fact authors have responded and made the distinction, AC felt the overall good merits of the paper slightly weigh over the concerning part.

Please incorporate all the revisions, esp. the comparison with recent SOTAs, missing references into the final version.